# Ensuring Urban Food Security in Malaysia during the COVID-19 Pandemic—Is Urban Farming the Answer? A Review

**Rosmah Murdad** [1,*], **Mardiana Muhiddin** [1], **Wan Hurani Osman** [2], **Nor Elliza Tajidin** [3], **Zainol Haida** [3], **Azwan Awang** [1] and **Mohamadu Boyie Jalloh** [1]

1   Faculty of Sustainable Agriculture, Universiti Malaysia Sabah, Locked Bag No. 3, Sandakan 90509, Sabah, Malaysia; mardianamuhiddin@gmail.com (M.M.); azwang@ums.edu.my (A.A.); mbjalloh@ums.edu.my (M.B.J.)
2   Centre for the Promotion of Knowledge and Language Learning, Universiti Malaysia Sabah, Kota Kinabalu 88400, Sabah, Malaysia; wanosman@ums.edu.my
3   Department of Crop Science, Faculty of Agriculture, Universiti Putra Malaysia, Serdang 43400, Selangor, Malaysia; elliza.tajidin@upm.edu.my (N.E.T.); haidacrzainol@gmail.com (Z.H.)
*   Correspondence: rosmahm@ums.edu.my

**Abstract:** Urbanisation and related insufficiency of food sources is due to the high urban population, insufficient urban food sources, and inability of some urban communities to afford food due to rising costs. Food supply can also be jeopardised by natural and man-made disasters, such as warfare, pandemics, or any other calamities which result in the destruction of crop fields and disruption of food distribution. The COVID-19 pandemic exposed the impact of such calamities on the fresh food supply chain in Malaysia, especially when the Movement Control Order (MCO) policy was first implemented. The resulting panic buying caused some food shortage, while more importantly, the fresh food supply chain was severely disrupted, especially in urban areas, in the early stages of implementation. In this regard, urban farming, while a simple concept, can have a significant impact in terms of securing food sources for urban households. It has been used in several countries such as Canada, The Netherlands, and Singapore to ensure a continuous food supply. This paper thus attempted to review how the pandemic has affected Malaysian participation in urban farming and, in relation to that, the acceptance of urban farming in Malaysia and the initiatives and approaches of local governmental and non-governmental organisations in encouraging the urban community to participate in urban farming through peer-reviewed journal articles and other articles related to urban agriculture using the ROSES protocol. About 93 articles were selected after screening to ensure that the articles were related to the study. During the COVID-19 pandemic, the surge in Malaysians' awareness of the importance of urban farming has offered great opportunities for the government to encourage more Malaysian urban communities to participate in urban farming activities. Limitations such as relevant knowledge, area, and space, however, are impediments to urban communities' participation in these activities. Government initiatives, such as the Urban Community Garden Policy (Dasar Kebun Komuniti Bandar (DKKB)), are still inadequate as some issues are still not addressed. Permanent Food Production Parks (TKPM) and technology-driven practices are seen as possible solutions to the primary problem of land and space. Additionally, relevant stakeholders play a crucial role in disseminating relevant and appropriate knowledge and methodology applicable for urban farming. Partnerships between government agencies, the education sector, and the private sector are necessary to develop modern urban agricultural technologies as well as knowledge, knowhow, and supports to build and sustain urban community participation in urban farming activities.

**Keywords:** hometown farming; Permanent Food Production Parks (TKPM); community gardens; Movement Control Order (MCO); vertical farming; hydroponics

## 1. Introduction

The ever-increasing annual migration from rural to urban areas has contributed to a steady increase in the urban population all around the world. It is estimated, however, that 32.7% of the global population in urban areas live in slums and over half of them live below the poverty line, such as in Angola, Bolivia, and Malawi [1]. In Malaysia, the urban population increased from 34.3% in 1971 to 77.2% in 2020 [2] due to increasing rural-urban migration, immigration, formation of new townships, and expansion of urban boundaries [3].

Generally, urbanisation implies an increase in the population density and results in unwanted consequences such as insufficient food sources and the inability of some urban communities to afford fresh food, especially fruits and vegetables, due to rising costs [4], as was experienced, for example, during the economic crisis in 2008. The decline in the growth of agricultural production caused a shortage in agricultural produce, resulting in increased global food prices in 2008 [5]. During that crisis, Malaysia faced a negative Balance of Trade (BOT) for food supply, such as for livestock and dairy products, fruits and vegetables, and rice, resulting in an undersupply of these items [5]. Appropriate measures thus need to be put in place to avoid a similar crisis in the future [6]. Food supply can also be jeopardised by natural (flood and landslide) and man-made disasters, such as warfare, pandemics, or any other calamities which result in the destruction of crop fields and disruption of food distribution, over a certain period. For example, at the end of 2019, the novel coronavirus disease (COVID-19) spread worldwide [7], with resultant unavoidable restrictions in movement, transport, and logistics. All of these factors further burdened city dwellers as it impeded their access to food supply and the distribution of agricultural produce to urban areas, leading to short-term shortages in food supplies.

COVID-19 also affected job opportunities and livelihoods in urban areas, especially amongst those involved in urban production, input supply, and marketing activities. People could not engage in normal daily activities, which affected their lifestyles and mental health [8,9]. Many were afraid to venture out of their homes due to the virulence of the virus. During the early days of the pandemic, the Malaysian government implemented a strict Movement Control Order (MCO) and as a result panic buying of food supplies caused some food shortages, and food supply distribution was also disrupted.

This situation has created increased awareness among Malaysians of the importance of producing their own food. This is evidenced by the drastic increase in participation in urban agriculture from 18,687 in 2019 to 40,219 in 2020 [10].

The most common approach in many countries is urban farming or hometown farming, such as in Canada, where urban farming is incorporated as a permanent land use of municipal parks, for extensive community garden programs. In the Netherlands, the urban farming areas are combined with several other lands uses in heavily populated areas. Meanwhile, in Singapore, the government promotes rooftop farms to overcome the limited land issue [11].

Urban farming can provide benefits in three aspects—environmental, social, and economic. Other than supplying food to the locals, urban farming can reduce import dependency, create job opportunities, and create a more sustainable environment. Urban farming can be one of the approaches to achieving sustainable agriculture [12] and food security for urban dwellers. In this respect it can be one of the major strategies to ensure that each household can secure some supply of fresh food. In Brazil, the United States, and several countries in Africa, urban farming is included as an important element in their food security strategies, and is promoted as an avenue to feed the citizens and give them access to healthy and fresh food [13,14].

In contrast, participation of Malaysian urban communities in urban farming activities before the pandemic COVID-19 era was not high. Lack of support from relevant stakeholders was one of the factors that made urban farming an unpopular activity [15]—this, however, is no longer the case, as during the recent pandemic period participation was seen to increase.

This paper attempts to review the acceptance of urban farming in Malaysia, the initiatives and approaches of local governmental and non-governmental organisations to encourage the urban community to participate in urban farming, and how the pandemic has affected Malaysian participation in urban farming. We hope that this paper will provide helpful information about the urban farming status, challenges, and future in Malaysia, and also its potential in ensuring food security, particularly in urban areas, during a crisis, such as the COVID-19 pandemic.

## 2. Methodology

### 2.1. The Review Protocol—ROSES

This study was guided by the Reporting standards for Systematic Evidence Syntheses (ROSES) review protocol. Its aim is to prompt researchers to ensure that they offer the right information with the correct level of details and was initially designed for systematic review and mapping for the environment management field [16]. There are five steps included in this study, from searching related articles to data synthesis and presentation. Figure 1 below shows the selection process for articles used in this paper.

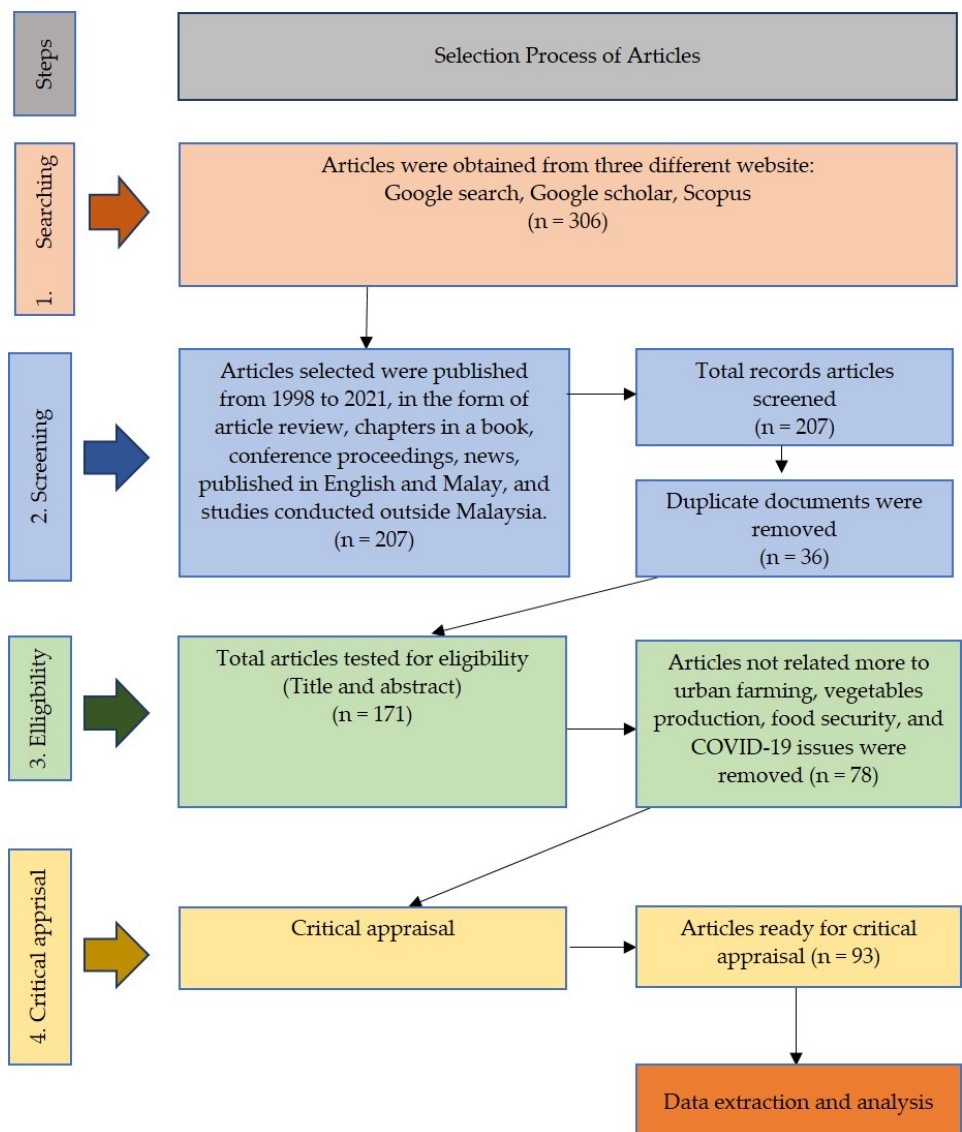

**Figure 1.** Selection process for articles [16]. The different colors in the flow chart indicate the various stages of evaluation and selection conducted for this article.

## 2.2. Searching

Peer-reviewed journal articles with a focus on urban farming in Malaysia were included in this review. Journal articles from 1998 to 2021 were first identified using the google scholar search engine. There were two languages used when searching the related articles: the English and Malay languages. Terms that were included for the search were: 'food security', 'Urban farming', 'pertanian bandar', 'kempen pertanian bandar', 'Malaysia', 'community perception', 'COVID-19 outbreak', 'vegetables import', 'food security', 'food availability', 'food security situations', 'socio-economic effect', and others. While there are many articles related to the terms used, only articles that were related to this study were chosen and identified during the screening process. Table 1 below shows the search string used in different websites for finding articles.

**Table 1.** The search strings.

| Database | Search String |
|---|---|
| Google search | 'food security', 'food availability', 'Urban farming', 'pertanian bandar', 'kempen pertanian bandar', 'Malaysia', 'community perception', 'community adaptation', 'community participation, 'COVID-19 outbreak', 'vegetables import', 'vegetables dumping, 'food security', 'food availability', 'food security situations', 'socio-economic effect' |
| Google Scholar | 'food security', 'food availability', 'Urban farming', 'pertanian bandar', 'kempen pertanian bandar', 'Malaysia', 'community perception', 'community adaptation', 'community participation, 'COVID-19 outbreak', 'vegetables import', 'vegetables dumping, 'food security', 'food availability', 'food security situations', 'socio-economic effect', 'pdf', 'article' |
| Scopus | 'food security', 'food availability', 'Urban farming', 'pertanian bandar', 'kempen pertanian bandar', 'Malaysia', 'community perception', 'community adaptation', 'community participation, 'COVID-19 outbreak', 'vegetables import', 'vegetables dumping, 'food security', 'food availability', 'food security situations', 'socio-economic effect' |

The words were typed with the quotation marks symbol ("-") to ensure that the results from searching the articles were focused on the terms used.

## 2.3. Article Screening, Eligibility and Critical Appraisal

References from the initially retrieved articles were examined, leading to additional articles being identified and retrieved. About 300 articles were retrieved before screening was applied. The screening stage was applied to remove any duplicate articles. During the screening process, criteria such as urban farming in Malaysia, Malaysian community perceptions on urban farming, COVID-19 effects on Malaysia's food security, and situations during the pandemic were focused on. It is impossible to review all existing published articles; thus, the timeline period of the articles was decided on to make it easier to review and to acquire more updated facts. Thus, in this article, we set the time period from 2001 to 2021, where urban farming had already begun and the COVID-19 pandemic happened.

After the screening process, 106 articles from the identified articles, not including articles from websites, were chosen to be reviewed. The articles were examined once again to identify the focus of the studies by reading the titles and abstracts. This was to ensure the eligibility of the articles in line with the criteria set. Overall, 106 articles were chosen as they fit the criteria, not including articles from websites.

## 2.4. Data Extraction, Synthesis and Presentation

In data extraction, the repeated or same terms used in articles were identified to compare the results. This is a mixed-method data presentation. The data were synthesised into simpler sentences by combining the results and some are presented using illustrations such as bar charts, tables, and figures. However, some of the data are presented in sentences. The data were extracted according to their themes, such as urban farming in Malaysia,

perception of urban farming, food security in Malaysia, and COVID-19 effects on food availability in Malaysia.

## 3. Results

### 3.1. Food Security in Malaysia before and during the COVID-19 Pandemic

The Food and Agriculture Organization (FAO) defines food security as "all people, at all times, have physical, social, and economic access to sufficient, safe, and nutritious food that meets their food preferences and dietary needs for an active and healthy life". Meanwhile, food insecurity was defined as limited availability or ability to acquire nutritional and safe foods in socially acceptable ways. Food insecurity is usually associated with poverty and undesirable health [17].

Globally, food security has been hit by significant yet uncertain impacts due to climate change, increasing populations, rising food prices, and environmental stressors. The focus of the International Food Policy Research Institute (IFPRI), in working on food security, includes analysing cash transfers, the promotion of sustainable agricultural technologies, and managing trade-offs in food security such as by balancing the nutritional benefits of meat against the ecological costs of its production. However, in addition to these measures, there is also a requirement for new adaptation strategies and policies which are targeted at the at-risk urban population. This includes plans for water supply handling, management of land use, food processing and safety, trade, and pricing, all targeted at ensuring continued food security.

In Malaysia, the national poverty rate was reduced drastically from 1970 to 2002, from 52.4% to 5.1%. While this is indicative of a reduction in the total number of poor households, it in no way shows that food security issues do not still exist, as emerged in the early days of the recent COVID-19 pandemic, which showed that the urban population in particular faced dire food supply situations. Further, it was found that, in general, Malaysian adults reduced their meal sizes and skipped main meals at least one or two months in a year due to financial constraints and the regularity of such problems was higher in East Malaysia (20.3%) compared with Peninsular Malaysia (11.5%) [18]. A study using findings from the Malaysian Adult Nutrition Surveys (MANS) in 2014 showed that 15.2% to 25.5% of adults experienced food quantity and variety insufficiency, small size of meals, and skipped meals due to financial constraints [17]. Figure 2 below shows the results based on an article about food insecurity in Malaysia.

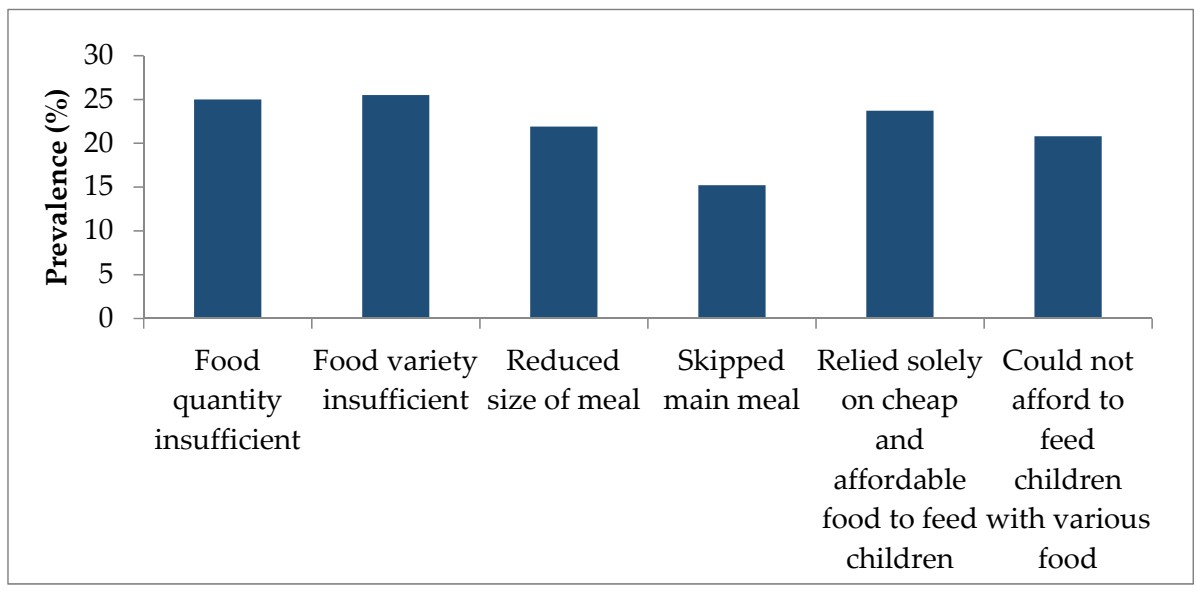

**Figure 2.** Prevalence of six parameters of food insecurity among Malaysian adults [17].

Socio-demographic status, such as source of income, household type, immigrant status, and marital status can have an impact on household food insecurity. A major factor associated significantly with food insecurity is household income [19]. This statement is supported by a study in Sabak Bernam, which showed that food-insecure households lacked dietary diversity, as the poor can only afford limited and cheaper food choices [20].

During the pandemic, the implementation of the MCO policy, which was first imposed in March 2020 to control the spread of the virus, caused limitations in food availability and accessibility by the people due to movement restrictions. The policy caused a decrease in domestic market activity in Malaysia, and in Sabah, during phase one of the MCO implementation, the distribution of vegetables from Kundasang to the nearest town was restricted and inadequate, resulting in dumping, as the sellers were unable to sell all their produce [21,22]. This restriction not only occurred within Malaysia, but also globally, as Malaysian import demand for vegetables such as onions, chillies, and leeks from China, Taiwan, and Thailand showed declines due to shipping restrictions.

Immediate measures thus need to be put in place to reduce the prevalence of such food insecurity issues before they become worse due to the ever-growing urban population and low income of households, especially in urban areas.

### 3.2. Malaysia's Socio-Economic Status during the COVID-19 Pandemic

On 10 April 2020, the Director General of the World Health Organization (WHO), Dr. Tedros Adhanom Ghebreyesus, revealed that the COVID-19 outbreak had affected 213 countries. The outbreak also affected the world food supply and distribution as food exports and imports were temporarily suspended. In Malaysia, the first COVID-19 positive case was reported on 25 January 2020, an imported case from Wuhan, China [23], and on 3 February 2020, the first Malaysian tested positive after travelling to a neighbouring country for a business meeting. On 11 March 2020, the situation worsened as a positive case from Brunei was traced to a religious gathering at the Seri Petaling Mosque in Selangor, sparking a new cluster of infections.

As a consequence of the increasing COVID-19 infections and mortality rates, varied strategies were put in place by the Malaysian Ministry of Health (MoH). One of the strategies which had adverse consequences for food security was the MCO implemented on 18 March 2020. While the objective of the restrictions in the movement of the populace was to control the virus spread, the stop order affecting all non-essential business premises and services led to local food distribution activities being curtailed and a resultant decrease in availability of fresh food supply, particularly in the urban areas. This was due to most of the urban fresh food sources being from outside of the urban areas.

The COVID-19 pandemic disrupted the public's health, economy, and daily life and 95% of respondents from a survey in Malaysia reported changes in their behaviour as they started to use personal-protective equipment such as face masks and gloves [24,25]. Furthermore, the disruption caused by this pandemic led to the sudden closure of business premises, panic, and increased anxiety in some places [26].

Most companies took the initiative of having their employees work from home, and some had to lay off a number of their workers. Among these, the 18–24 age group of Malaysian respondents were reported as experiencing higher job losses, which comprised 42% of the total respondents [27]. In some food retail sectors, the number of workers decreased, thus affecting food production quantity and subsequent limitations in food supplies. Furthermore, in some areas, such as in Sabah, fresh vegetable supplies were dumped as the vegetables could not be distributed due to the implementation of the MCO policy [28]. Additionally, a study on the impact of COVID-19 on the wellbeing of older persons in Malaysia reported that 35.3% of the respondents experienced 20% to 50% reduction in their incomes, while 37.1% of the respondents suffered a reduction of more than 50% of their original income [27]. Figure 3 below shows the impact of COVID-19 on household income (%) in Malaysia.

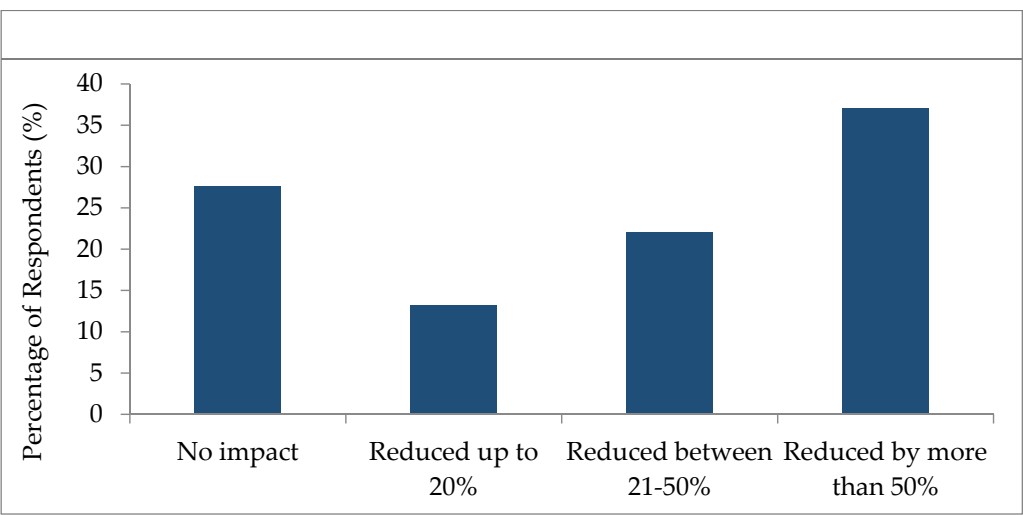

**Figure 3.** Impact on Household Income (%) [27].

The Malaysian government and also NGOs have been actively helping those affected by the pandemic in various ways, such as by providing protective masks and shelter for the homeless. Further, on 27 March 2020, the Prime Minister announced the PRIHATIN Package, valued at RM 250 billion, to support micro- to medium-sized enterprises, and an additional budget of RM 1 billion for medical needs [29].

### 3.2.1. Economic Situations in Food Marketing and Distribution

During the MCO implementation period, Malaysia suffered losses of up to RM 2.4 billion a day, with an accumulated loss of RM 63 billion up to the end of April, in terms of the economic impact. The sudden enforcement of the MCO put the economy of various sectors in jeopardy, including the agriculture sector [30]. The Food supply and distribution chains were among the most adversely affected due to this measure. The consequent far-reaching impacts were the aggravation of poverty and food security. The poor and vulnerable households, particularly larger households and those with children, felt the effects the most as many of the breadwinners of these households also faced income loss during this period.

Approximately 68% of farmers lost their income and capital, although a majority of them continued to operate, and 91% faced difficulties in selling their agricultural products as the MCO policy disrupted the market channels. However, the distribution and marketing channels were more severely affected during the implementation of the MCO as compared with during the Conditional Movement Control Order (CMCO). Figure 4 below shows the difficulty in distribution and marketing faced by economic clusters during the MCO and CMCO implementation periods. Meanwhile, Figure 5 below shows details of the distribution and marketing problems faced during the MCO and CMCO implementation. A distinct change was also seen during this period, as supply chain disruptions resulted in farmers who were unable to sell their produce in the market instead sharing their produce with neighbours and relatives, thus alleviating their situations. This scenario, however, was mostly limited to the rural areas, while those in urban areas continued to suffer from lack of supply. The urbanites were also constrained in their ability to grow their own food due to limited land and knowledge, among other factors.

The MCO and CMCO also affected the food intake of Malaysians whereby the food distribution problems were more severe during the CMCO period as compared with those during the MCO period. A study conducted in four selected states in Peninsular Malaysia, Selangor, Perak, Kelantan, and Johor, to represent the central, northern, eastern, and southern regions in Malaysia, respectively, showed that food insecurity and insufficiencies of protein and fibre consumption increased during the COVID-19 pandemic [31]. They also concluded that the food insecurity and insufficiencies of protein and fibre intake due to

the implementation of the MCO may also be associated with consequential psychological distress among Malaysians, particularly the middle-aged and older adults [31].

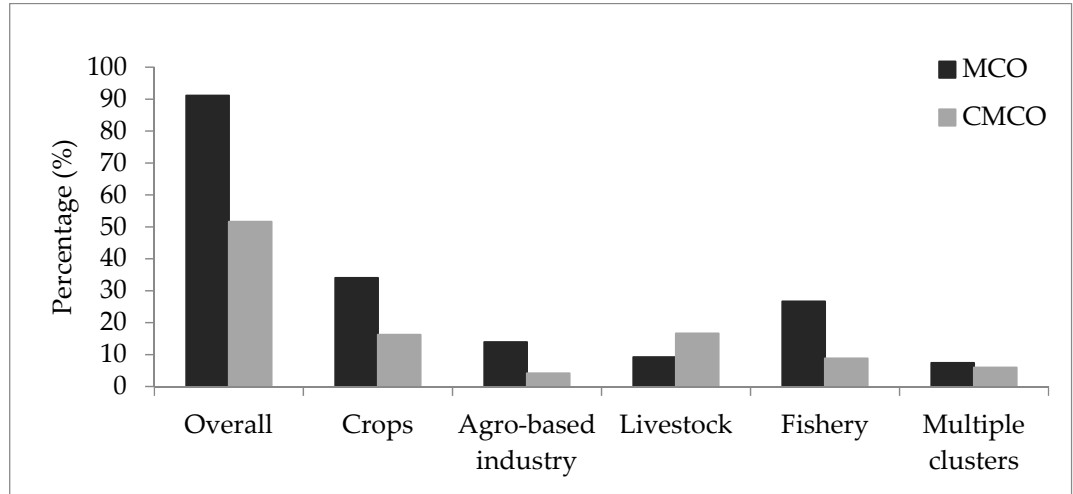

**Figure 4.** Difficulty in distribution and marketing by cluster during MCO and CMCO [30].

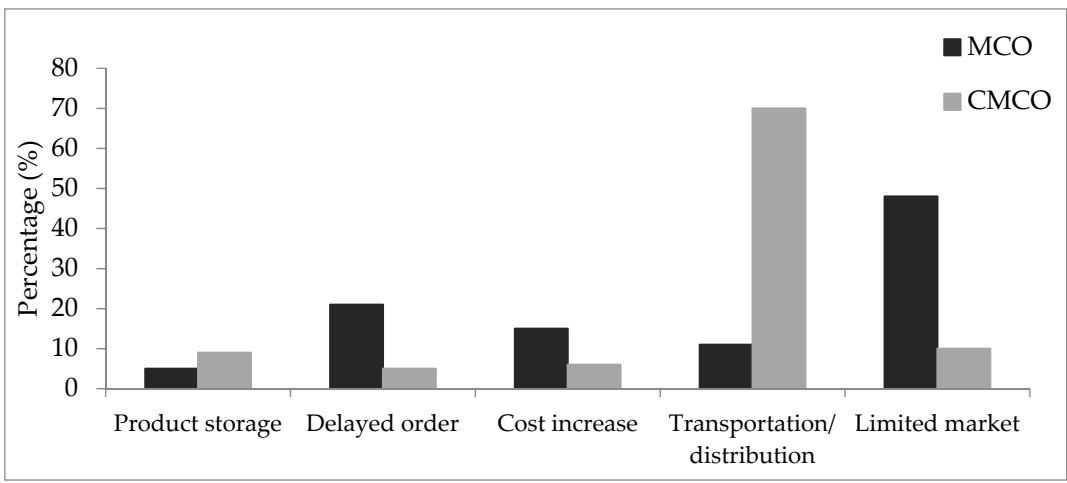

**Figure 5.** Details of distribution and marketing problems during MCO and CMCO [30].

### 3.2.2. Government Strategies and Initiatives to Encourage Community Participation in Urban Farming

In Malaysia, fresh food supply is generally sourced from local food producers and supplemented by imported food. About 20% to 30% of the annual Malaysian food supply is imported, to meet the needs of the Malaysian population [32]. The COVID-19 pandemic outbreak had a big impact on the economy, displayed by a decline in export and import demand, and a decrease in domestic market demand, primarily due to the impact of the implementation of the MCO policy. A major portion of Malaysia's vegetable supply comes from abroad, including onions, chillies, and leeks, with China, Taiwan, and Thailand being among the main suppliers. Malaysia's balance of trade for vegetables is very much skewed, with imports outstripping exports by almost five folds, where the 2016 total import value was RM 5,073,905, and exports only RM 1,504,883 [33]. However, in 2020, the import and export values decreased by 4.36% and 11.4%, respectively, compared with those in 2019 [33].

Cities in Malaysia rely on food supply from farms that are normally located far from the cities, whereby the food has to be transported over land. During the COVID-19 pandemic, the food supply chains were disrupted, mainly in urban areas, due to the

MCO policies [34]. Additionally, farmers were forced to give away or dump their farm produce due to its perishable nature as they were unable to sell it, especially during the first lockdown period [35]. Consumers such as foreign workers, refugees, and the lower-income groups were consequently unable to fulfil their daily dietary requirements due to difficulties in accessing food during the MCO [32].

During MCO 1.0, which spanned from 18 March 2020 until 3 May 2020, 63 controlled fresh market locations were opened to ensure that the public had access to fresh food sources. After the MCO 1.0, the government improved their strategies to ensure adequate supply of food for the community by allowing selected sectors to operate, such as the agricultural, fishery, livestock, plantation, and commodity sectors, subject to approved standard operating procedures (SOP) being put in place. However, when the number of COVID-19 positive cases in Malaysia kept increasing, the government implemented another lockdown in June 2021, to curb the increase. Due to the implementation of the SOPs during the lockdown, the Minister of Agriculture and Food Industries, Datuk Seri Dr Ronald Kiandee, stated in May 2021 that only 43 controlled fresh market locations nationwide, managed by the Federal Agricultural Marketing Authority (FAMA), would be open for public access, to ensure food supply needs were met [36].

The Agriculture and Food Industry Ministry set aside an allocation of RM 10 million under the National Economic Recovery Plan (Pakej Jana Semula Ekonomi Negara (PEJANA)) initiative, to be distributed to 800 communities and 12,000 new participants joining the urban farming programme. The B40 group in particular was encouraged to participate in the programme as a group so that they could grow their own food. B40 stands for 'Bottom 40%', and Malaysians within the B40 group are those with median monthly household incomes of RM 3000 or below RM 3860.00 per household. The PEJANA incentive was to assist existing and new participants by providing them agricultural inputs in the form of seeds, fertilisers, infrastructure, tools, and advisory services and training [37].

The government also launched the Community Garden Programme (Program Kebun Komuniti) to encourage Malaysians to take up urban farming. The launch of this programme helped participants to reduce their grocery expenses and secure food sources [37]. Some of them even shared their produce with neighbours.

On 3 August 2021, Datuk Hjh. Zuraidah binti Kamaruddin, the Minister of Housing and Local Government, launched the Urban Community Garden Policy (Dasar Kebun Komuniti Bandar (DKKB)) to encourage urban communities to become involved in urban farming (UF) activities.

### 3.3. Urban Farming

Urban farming is defined as the production of crop and livestock goods within cities and towns, while peri-urban agriculture areas around cities and towns can also be considered as UF as they provide products to the local population [38–40]. It involves the production of crops and livestock in the home or in plots in urban or peri-urban areas [4]. The term 'urban farming' is used to describe both plant cultivation and animal rearing for home consumption and income generation in cities. Urban farming also includes other interrelated activities, such as the production and sale of agricultural inputs, and postharvest handling and marketing of agricultural produce [4].

Even though agriculture has always been associated with a rural environment, the urbanisation process where rural areas are converted into urban areas has affected the agricultural system. In Brazil, this has resulted in diversity of vegetation types within 170 home gardens and 98 plant species identified in 21 urban gardens, including a large diversity of fruit trees and shrubs [40]. Furthermore, the cultivation activities also varied from the planting of vegetables, herbs, mushrooms, fruit trees, and ornamental plants to keeping livestock for eggs, meat, or dairy products.

### 3.3.1. Types of Urban Farming

There are many types of UF, such as community gardens, private gardens, easement gardens, roof-top gardens, and urban orchards (Table 2). The practice, referred to as Zero-acreage farming or ZFarming, is where urban buildings are utilised for agricultural activities. This generally involves three types of activities, i.e., rooftop gardens/rooftop farms, rooftop greenhouses, and indoor farms. ZFarming practice is highly versatile and can be used for both commercial and non-commercial production. In relation to food security for urbanites, Zfarming can contribute to increased food supply of high quality and safety for the local population, where the need to depend on transportation is also reduced. It can contribute to developing sustainable supply for the urban population. Definitions of each type of UF are shown in Table 2. As clearly shown by these definitions, UF does not just contribute to ensuring a steady food supply for the urbanites. It can also contribute to improved community relations and also the environment. As a matter of fact, UF can contribute positive impacts for city dwellers, whether social, environmental, or economic [4].

**Table 2.** Definition of different types of UF (commercial and non-commercial).

| Types of UF | Definition | References |
|---|---|---|
| Community garden | Represent small-scale, highly patchy and qualitatively rich semi-natural ecosystems; usually located in urban or semi-urban areas for food production; to improve social, economic, cultural, and environmental conditions. | [41–44] |
| Private garden | Located in suburban areas and the most prevalent form of UF in cities | [45] |
| Easement garden | Located within community properties but often regulated by the local government; established to improve water quality and for erosion control. | [46,47] |
| Roof-top Garden | Any garden established on the roof of a building (for decorative or agricultural purposes) | [48] |
| Urban orchard | Tree-based food production systems that can be owned or run privately/by the community. | [49] |
| Institutional garden | Located at any government or non-governmental institution/buildings which are managed by an organisation; not necessarily for food production | [50] |
| Demonstration garden | Small gardens located in private areas, housing areas, and commercial areas for demonstration purposes only; referred to as learning or teaching gardens. | [51] |
| Guerrilla garden | Ornamental/crop plants onabandoned/unauthorised land, used for neighbourhood improvement. | [46] |

### 3.3.2. Benefits of Urban Farming/Hometown Farming

Urban Farming can have positive impacts on various adverse situations in urban settings. However, if it is poorly managed, it may cause health and environment problems.

The health benefits associated with UF include increasing households' access to healthy food, increased vegetable and fruit consumption, increased physical activity, improved mental health, and improved quality of life and general well-being. In China, the households that relied on self-cultivated food environments did not experience changes in their diet during the pandemic compared to households that were primarily reliant on formal built food environments. This shows that households in urban communities are able to secure their food sources by farming. In addition, besides providing nutritious, healthy,

and affordable food for the urban communities, UF can also improve the livelihoods of the households [52,53], as vegetable gardens in home spaces provide recreation, enhance physical and mental health, and provide economic and environmental benefits [54]. A readily available and easily accessible healthy fresh food source within the home vicinity is likely to enhance healthful dietary intake among youth and families [55].

Urban farming is said to have social benefits by improving the community food security, provision of educational facilities, and serving as a design inspiration. Economy wise, it helps to provide potential public benefits and commodity outputs [56]. Urban farming can improve food security in low-income communities as it can be applied anywhere, on small-scale or large-scale. Not all apartments have large terraces or courtyards to devote to the cultivation of vegetables; often, the area is limited to a balcony and windowsill, where these spaces can still be used to successfully cultivate a small vegetable garden [54]. Table 3 below lists the benefits and potential of UF.

**Table 3.** Four Benefits and Potential of Urban Farming.

| Benefit Categories | References |
| --- | --- |
| (a) Environmental Stewardship<br>Urban revitalisation<br>Ecosystem enhancement<br>Climate change<br>Waste reduction<br>Sound insulation<br>Noise absorption<br>Minimize transport footprints<br>Enrich visual quality | [57–60] |
| (b) Social Improvement<br>Community empowerment<br>Youth development and training<br>Green education opportunities<br>Cultural integration and preservation<br>Positive community interaction and social well-being | [61–65] |
| (c) Health and Nutrition<br>Food access and security<br>Organic fruits and vegetables<br>Nutrient retention<br>Therapeutic treatment<br>recreational and garden space amenities<br>Physical activities and exercise<br>Aesthetic values | [55,63,66–69] |
| (d) Economic Reliance<br>Cash-free nutrients<br>Lowering expenditure<br>Income generator<br>Green innovation<br>Employment creation<br>Introduce local food production<br>Positive economic description | [57,62,63,66,69] |

Urban farming can provide universal benefits towards communities in achieving sustainable environmental and social improvements, economic reliance, and health and nutrition improvement, and also plays an important role in improving the community's mental and physical health [70–74].

Urban farming may, however, have negative health implications if vegetables are grown in soil with high lead concentrations, which is a major problem. Lead or any heavy metal content can result in health issues. It is thus important that urban soils are properly managed to reduce their Pb (lead) content and any subsequent uptake by food crops grown

on them. In a relevant study, it was shown that vegetable consumption was the main contributor for the excess of provisional total tolerable intake (PTTI) levels of Pb in 40% of children and 10% of gardeners. The authors recommended that healthy gardening practices be carried out in urban community gardens to reduce the risks of Pb exposure. They further recommended that lower Pb vegetables be cultivated and also for the root-zone bed soil in raised beds to be frequently replaced with clean soil [75].

Caution is also advised to avoid environmental pollution due to excessive use of insecticides in urban farms and also to guard against women who participate in such activities being overworked. Care also needs to be taken in the choice of system used in UF to avoid incurring high costs [76].

*3.4. Status of Urban Farming in Malaysia and Perception of the Community*

Major cities in Malaysia such as Kuala Lumpur, George Town, Petaling Jaya, Kuching, Miri, and Kota Kinabalu show an increasing urban population growth trend [53]. The growth of the Malaysian urban population is expected to continue to increase and, thus, the food supply demands will also increase, which was estimated at 70% in 2020 [77]. This will cause the urban communities to face higher living costs due to the rising costs of food production, processing, and distribution. In view of the impending phenomenal increase in food supply demand, the government is committed to ensuring that the food security of communities is satisfactory and has thus introduced a variety of methods, technologies, and innovations targeted at increasing productivity.

Urban agriculture, which covers all agriculture-related activities in urban areas, is the urban communities' attempt to supplement their food needs via self-production. This was further proven through a survey which found that 41% of the respondents engaged in such activities did so for self-consumption [78]. It can undoubtedly alleviate the ever-rising daily expenses and cost of living in urban areas. Urban communities are seen to be increasingly moving towards this practice for self-consumption, while some are also venturing into developing semi or fully commercial farms in urban areas [52].

3.4.1. Participation of the Malaysian Urban Community in Urban Farming Activities

After Malaysia achieved independence in 1957, the government developed strategic plans to achieve future self-sufficiency in food production through the National Agricultural Policy (NAP) [79]. In line with this, the Green Book Programme (Rancangan Buku Hijau) was initiated in 1974 by the then Prime Minister of Malaysia, Tun Abdul Razak Hussein. Despite the success of the Rancangan Buku Hijau in mitigating economic crisis and food security problems by reducing the inflation rate and increasing household incomes in the early 1980s, a new economic crisis emerged as Malaysia's overall export price index fell by 30% as tin and palm oil prices plunged, thus affecting the urban areas [61]. In 2006, the Green Earth Campaign was launched by Tun Abdullah Ahmad Badawi, the Prime Minister at the time, as an avenue to mitigate food security issues, the concept being to promote implementation of UF among urban residents to encourage them to grow their own food [61]. The promotional activities under this campaign as well as the Pembangunan Keluarga Tani (Farming Family Development) initiative encompassed UF. All of these were extensions of the Green Book programme [61]. The focus on UF was given an additional boost in 2013 with the establishment of the Urban Farming division in the Department of Agriculture. Additionally, in 2014, several programmes such as Community Farming (Pertanian Komuniti) and Jom Bertani were launched by the government, all designed to enhance awareness about UF amongst the local urbanites. More recently, in 2021, a new program called Urban Community Farm (Kebun Komuniti Bandar) was launched. The purpose was to encourage urban communities to produce their own food, especially during crisis periods.

In some urban areas in Malaysia, UF is being practiced to improve food security and nutritional intake. A study on UF impacts in 2017 showed that in Selangor, the UF production by one family helped reduce their expenses on vegetables by RM 124 per month,

while in Kuala Lumpur, participants saved RM 92 that was usually used to buy vegetables. Additionally, in Pulau Pinang, in a senior citizen care centre, they successfully produced a total of 5100 kg of vegetables in 2020 compared with only 4200 kg of vegetables in 2019.

Types of UF systems that can be implemented in areas with limited space or land availability include household gardens to grow and produce food items for family consumption; community gardens/farms—for production of food and neighbourhood beautification purposes; office gardening—corporations cultivating food on rooftops as they become more environmentally conscious; and school farms or gardens—usually for educational purposes [53]. Urban farming production systems such as integrated farming, various types of hydroponics, rain shelters, aeroponics, and organic farming are the major production systems observed in the Malaysian UF practice.

However, not all urbanites practice or become involved in UF activities in Malaysia. Eating out is a trend among urban workers, students, and even families in Malaysia. Having no time to cook or prepare food, working away from home, and being a working mother are among the reasons why the people like to eat out. Furthermore, the variety of food served at many premises encourages the practice of eating out [3]. Due to this trend, it is difficult to get the Malaysian urban communities involved in agricultural activities.

However, after the occurrence of the COVID-19 pandemic in 2020, the number of UF locations in Malaysia was seen to increase drastically, up to 17,320 locations compared with only 735 locations in 2019. Table 4 below shows the statistics of UF project locations in Malaysia.

**Table 4.** Number of urban farming locations in Malaysia (2014–2020). Courtesy of Urban Farming Division, Department of Agriculture Malaysia, Putrajaya.

| Category | 2014 | 2015 | 2016 | 2017 | 2018 | 2019 [a] | 2020 [b] | Total |
|---|---|---|---|---|---|---|---|---|
| Residential (Individual) | 63 | 379 | 427 | 100 | 195 | 92 | 16,005 | 17,261 |
| Residential (Community) | 87 | 124 | 247 | 445 | 190 | 156 | 820 | 2069 |
| Schools | 75 | 296 | 525 | 137 | 320 | 352 | 332 | 2037 |
| Institutions/Public/Private | 55 | 191 | 246 | 72 | 166 | 135 | 163 | 1028 |
| Total | 280 | 990 | 1445 | 754 | 871 | 735 | 17,320 | 22,395 |

[a] Before the COVID-19 pandemic; [b] During the COVID-19 pandemic.

Support from the government and related agencies has helped make UF more popular among the urban communities in Malaysia, especially in the low-cost residential areas. Among the participants in UF programmes from urban communities and peri-urban communities are retirees, housewives, teenagers, and working people. As seen in Table 5 below, the number of participants increased from 18,687 in 2019 to 40,219 participants in 2020.

**Table 5.** Participation in urban farming in Malaysia (2014–2020) [10]. Courtesy of Urban Farming Division, Department of Agriculture Malaysia, Putrajaya.

| Category | 2014 | 2015 | 2016 | 2017 | 2018 | 2019 [a] | 2020 [b] | Total |
|---|---|---|---|---|---|---|---|---|
| Residential (Individuals) | 682 | 10,731 | 11,088 | 2160 | 4319 | 1555 | 17,403 | 47,938 |
| Residential (Communities) | 1312 | 2646 | 5372 | 14,384 | 4731 | 4056 | 9981 | 42,482 |
| Schools | 1399 | 8215 | 11,913 | 3298 | 8412 | 9706 | 8709 | 51,652 |
| Institutions/Public/Private | 1609 | 3580 | 4582 | 1991 | 3877 | 3370 | 4126 | 23,135 |
| Total | 5002 | 25,172 | 32,955 | 21,833 | 21,339 | 18,687 | 40,219 | 165,207 |

[a] Before the COVID-19 pandemic; [b] During the COVID-19 pandemic.

As shown in Table 5, participation in UF activities has been unstable, as it increased until 2016 but started to decrease in 2017. However, in the year 2020, participation increased drastically, especially among individuals in residential areas. The onset of COVID-19 was a wake-up call which increased awareness of some communities of the importance of growing

their own food in order to secure their food sources, especially during the lockdown phase. The MCO restricted their movements especially in being able to purchase fresh food. Farming in their homes was a way of securing food availability to have a healthy diet. During the pandemic, a study showed a reduction in shopping frequency but not shopping locations, and most products purchased were for health reasons such as vegetables, and for mood improvement, such as chocolates [80]. This suggests that the pandemic situation affected peoples' behaviour in terms of where and what to shop for, as well as influenced what they felt a need to buy and consume.

In Malaysia, UF has been practised in a variety of scales, from small to medium [81]. The majority of the urban farmers carried out their activities on small areas of land around their house, or community type farming on medium-sized land areas in residential areas. Further, most of the vegetables cultivated in Malaysian urban farms are temperate and tropical vegetables, such as leafy vegetables (cabbage, spinach, lettuce) and even tomatoes and brinjal [52]. These vegetables were mostly grown in small gardens, vacant plots, balconies, and containers. The Department of Agriculture (DOA) usually helps new urban farmers by determining the areas of activity, type of category, and the suitable crops to be planted. Figure 6 below shows examples of UF practices in Malaysia.

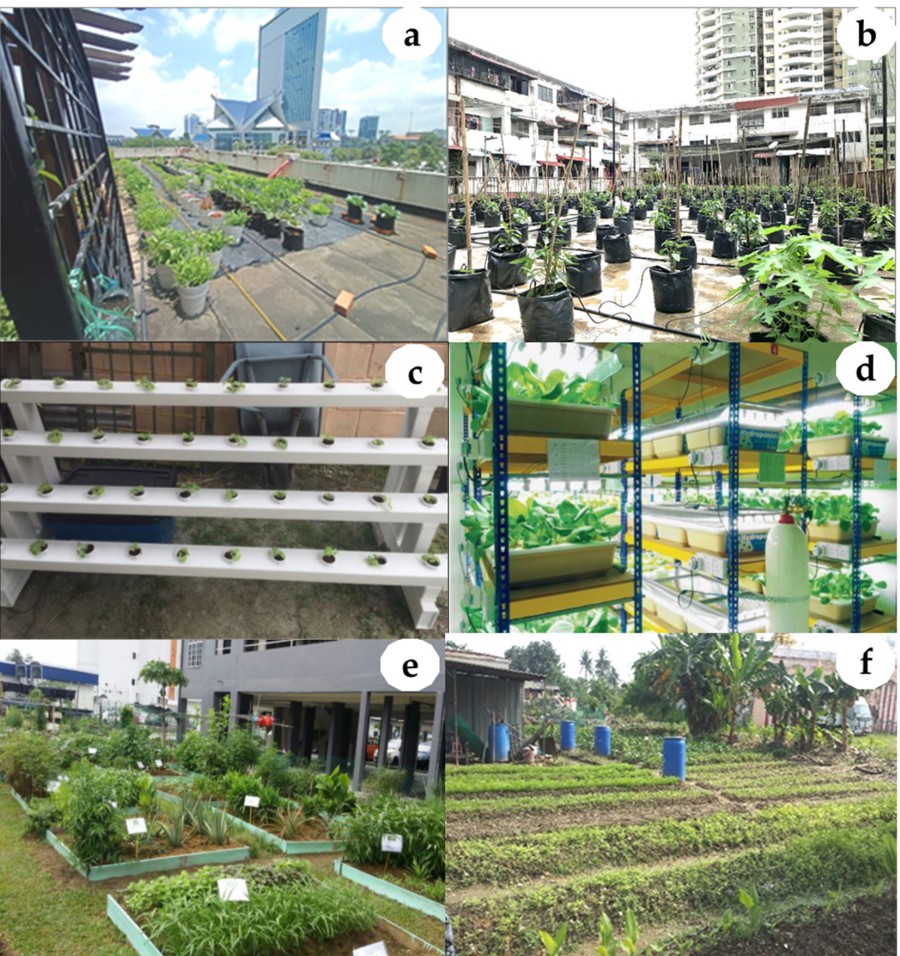

**Figure 6.** Urban farming in Malaysia: (**a**) rooftop farming at Department of Agriculture (DOA) Shah Alam, Selangor; (**b**) the Hijau Roof located at Brickfields, Kuala Lumpur-Utilisation of a rooftop parking lot for farming by a private company. Photograph courtesy of Mrs. Chong Khai Yeng; (**c**) Mini hydroponic system set-up by a resident; (**d**) plant factory—an indoor hydroponic system developed by the DOA at Gombak Selangor; (**e**) planterbox; and (**f**) conventional cultivation using raised beds at the Section 18 Community Farm (Kebun Komuniti Seksyen 18), Shah Alam, Selangor.

Figure 7 below shows different types of farming activities in urban areas in Malaysia, based on materials used. The range is from small areas of land around the house, to community type farming on medium-sized land in residential areas, and recycling of items into agricultural tools. Most of the farmers implemented conventional types of gardening without the need for high technology, such as hydroponic or fertigation systems, as they are more practical, incur low initial costs and do not require special skills or knowledge. Some urban farmers are also aware of the benefits of production of organic food crops using kitchen waste compost and natural farming practices, as shown in Figure 8.

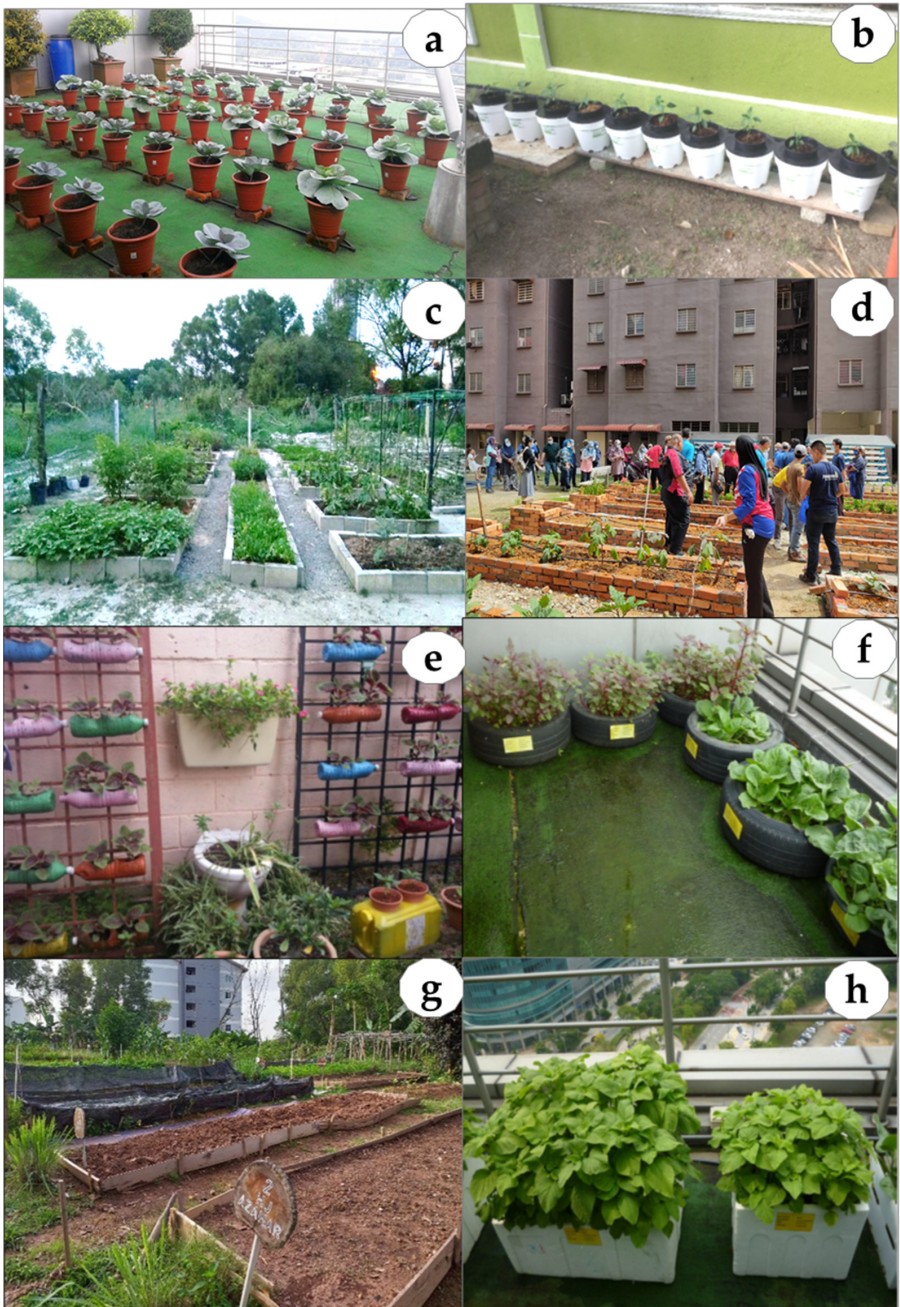

**Figure 7.** Different urban agriculture design ideas by Malaysians based on materials used. Photograph courtesy of Urban Farming Unit, Department of Agriculture, Selangor: (**a**) using pots; (**b**) using smart pots; (**c**) using concrete bricks; (**d**) using common burnt clay bricks; (**e**) using recycled plastic bottles; (**f**) using recycled tyres; (**g**) using reclaimed wood; (**h**) using styrofoam containers and the most common, polybags (not in the picture).

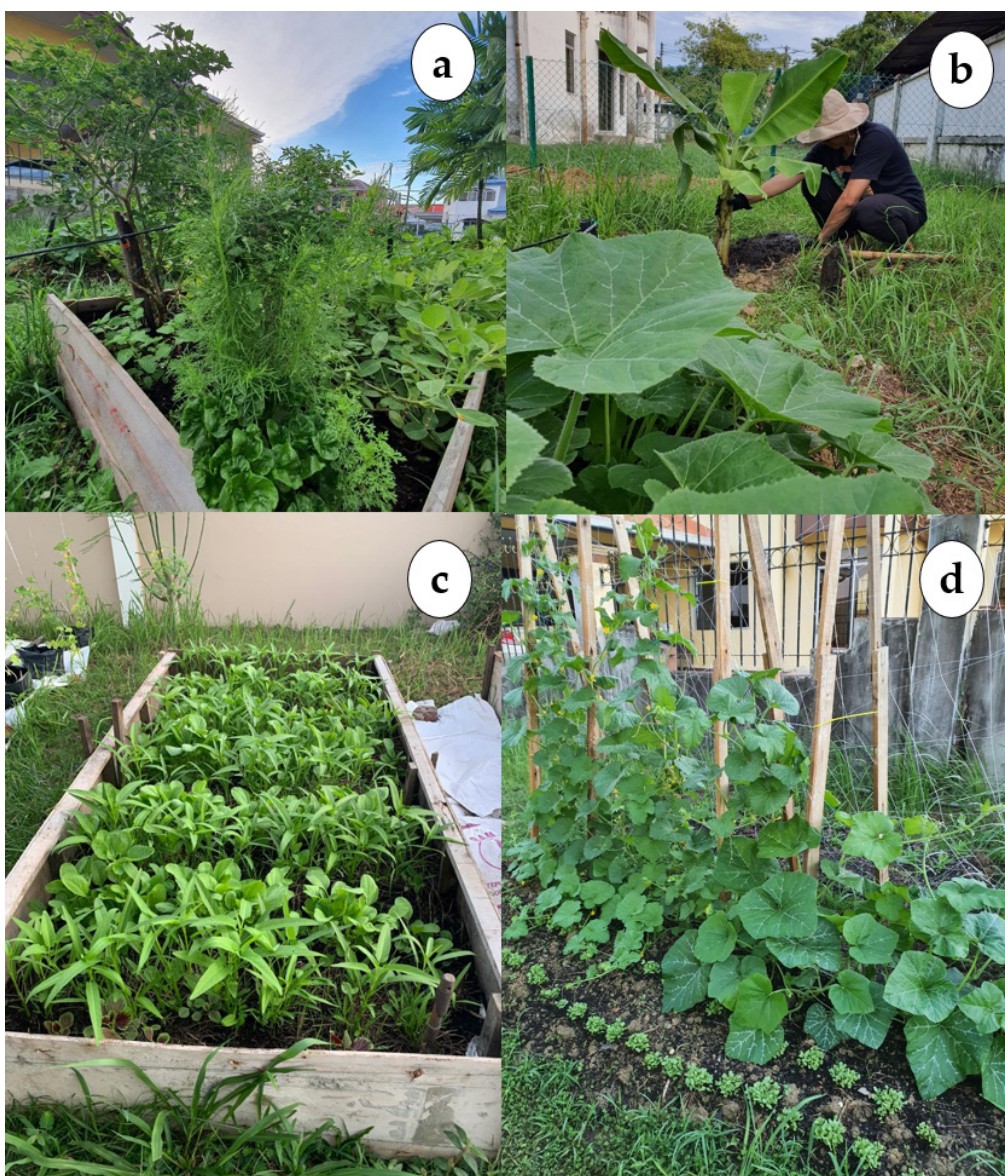

**Figure 8.** Organic urban farming by a resident in Taman Megah, Sandakan, Sabah, Malaysia. (**a**) varied vegetables in a single planter box; (**b**) banana plant and pumpkin; (**c**) water spinach and mustard intercropped; (**d**) cucumber. The farmer used kitchen waste compost—courtesy of Mr. Muhammad Shahrul Khalid.

### 3.4.2. Community Expectations from Urban Farming Activities

There are no significant studies which show how urban communities in Malaysia are participating and carrying out UF in their respective areas, especially in Sabah. However, acceptance of urban agriculture among communities in the city is influenced by the levels of cognitive, affective, and behavioural components of their attitudes [77]. There are two major motivators for communities to participate in UF, which are social and health benefits [58,60]. Research on the 'perception of community residents on supporting UF in Malaysia' found that there is full potential for residential communities to participate in urban farming activities by implementing community gardens, however it was difficult to sustain their interest for a long term.

In a related study, survey respondents expressed expectations that their participation in UF would present environmental benefits in terms of beautification, improved aesthetic value of the neighbourhood, and reduced pollution [82]. Malaysian urban farmers, on the other hand, expect their urban farming participation to enhance their physical and

mental health, and also provide social interaction benefits and fresh produce. This was supported by a survey study in 2018, wherein it was reported that the respondents expected their urban farming activities to provide them social, health, environmental, and economic benefits. However, the residents also expressed concerns about the soil quality as some soils were contaminated with lead (Pb), zinc, cadmium (Cd), and copper (Cu), but not arsenic (As) or chromium (Cr) [75].

In relation to youth participation in UF, a study showed that the strongest predictor of participation was students' attitudes toward urban agriculture, followed by subjective norms, career motives, and perceived barriers to participation [83]. It was mentioned that the younger generation and more educated individuals with a larger household size show a positive attitude towards urban agriculture activities [83]. Career motives were a significant predictor of students' intention to participate in the Universiti Putra Malaysia's (UPM) UF program. Providing 'career-aligned' students opportunities to volunteer in UF helped expand their knowledge on agriculture. Youths in Malaysia who come from rural backgrounds have little interest in traditional agriculture. Universities should thus take action by developing UF programmes that allow students to show their creativity, create jobs, and, lastly, are profitable [83].

### 3.5. Role of Stakeholders in Facilitating Urban Farming for Food Security

Urban farming is now gaining popularity and being practised by many individuals or communities in the country. It is therefore important to ensure that these farmers have adequate knowledge about UF to ensure that the objectives of UF are achieved. The contributions and impacts of UF on Malaysia's food security include good supply of adequate nutritional diet for the people, especially in urban areas. This has perked the interest of many stakeholders in Malaysia, including policymakers, practitioners, academics, and the private sector, which is crucial for its development. In Indonesia, for example, the development of UF in Jakarta required the support of sectoral stakeholders [84,85].

During the first phase, it is important for a UF farmer to acquire accurate and adequate information about agriculture. Here, the extension agent will play an important role in delivering the required knowledge, including the concepts, benefits, and methods of implementation. The delivery of knowledge can be implemented through extension programs such as via seminars, agriculture fairs, and others [77].

Different stakeholders can contribute different types of knowledge to deal with uncertainties and risks, and to contribute to planning and framework or design. There are three different approaches in stakeholders' roles in UF: (1) Stakeholder-based initiatives; (2) Government-based stakeholders; and (3) Science-based stakeholders.

As the Malaysian government has started to promote UF formally, selected agencies and public institutions such as UPM, the nation's premier public agricultural university, have been selected to carry out training and promotion of UF to urban residents [81]. Further, the 'Urban Agriculture' program was created with assistance provided by DOA and other selected stakeholders. The purpose of the program was to help households, particularly poor families, to reduce living costs [86].

In Malaysia, both governmental and non-governmental bodies, such as the DOA and UPM, play crucial roles in the promotion of UF to urban households, providing them relevant information and assistance [86]. Researchers from UPM further attest to the capability of UF in addressing food security issues, as it is believed to be able to provide urban communities access to nutritious and safe food and to save on food costs [86]. Meanwhile, a study at Universiti Teknologi Malaysia (UTM) on three major ethnic (Malay, Chinese, and Indian) urban residents found that their motivation in cultivating plants was for different purposes, including aesthetics, culinary, and spiritual purposes. This indicates that the education hub also has a role to play in the introduction of UF to the urban communities, especially in influencing the youth to participate in UF.

Furthermore, according to research on Predicting Youth Participation in Urban Agriculture in Malaysia, youth participation is influenced by students' attitudes towards UF,

subjective norms, career motives, and perceived barriers, while a related study mentioned that organisational support and work assignments are factors associated with students' voluntary participation [83]. Meanwhile, perceiving benefits, gaining experience, improving confidence and self-esteem, attaining emotional stability, and altruism are the motivational factors towards volunteerism among Malaysian youth [87,88]. Providing opportunities for student involvement in activities with communities off-campus can help students to experience the impact of UF themselves, as compared with what might be taught during lectures. In line with this, UPM plans to recruit undergraduate students to be trained as volunteers for UF programs being implemented throughout the nation. Their role would be as change agents, providing interested urban residents useful information on production, processing, and marketing in relation to UF [89]. It is also hoped that this will indirectly build up interest and involvement of youth in UF activities [83].

## 4. Discussion

At the rate that current development projects are progressing, it is foreseen that the urban environment will be even more dominated by built structures, either buildings or facilities. This will mean that green areas in urban environments will shrink, along with access and availability of fresh food sources, which will be all the more felt during times of crisis such as pandemics and natural calamities such as flooding and similar situations. As a developing country, Malaysia faces several challenges which make UF an important policy consideration, especially in terms of food safety and security. The increase in living costs for urban communities due to the rising costs of food production, processing, and distribution [38] has drawn the attention of Malaysian policymakers to seek new ways to offset the higher food costs and expenditure on imports, to enhance food security, reduce consumption of processed food, and other related issues.

Lack of knowledge on UF is one of the problems faced by the urban communities in Malaysia in starting UF activities in their areas. Insufficient extension education services and community outreach efforts to develop and provide services to the communities are among the major challenges faced by Malaysian urban dwellers [15], including lack of knowledge or education [77]. Other major challenges identified for UF activities in Malaysia were perceptions of limited space, resources, and education. Malaysia also faces scarcity of trained and knowledgeable personnel which severely hampers extension activities that are vital in promoting UF activities [53]. A study on UF as a way to achieve food and nutritional security showed that the younger generation is more flexible to adapting and accepting changes compared with the older generation, which is more cautious in dealing with new phenomena [86]. In addition, younger urban dwellers with higher levels of education are more likely to relate UF to food security.

Shortage of available land and access to water is another major problem facing UF activities in Malaysian urban centers. Due to limited land and space availability, maximising crop production per unit area is an important issue to be addressed when it comes to food sources for self-consumption, especially for households in urban areas [52]. For example, as land availability is one of the major problems in UF activities, the government needs to think of ways to expand agricultural land, particularly in urban areas, to ensure a stable supply of staple foods and the food security of the urban population in the future. The introduction of different types of UF systems such as hydroponics, vertical gardens, and rooftop gardens can solve the land availability and space problems.

Credible and suitable extension services would also play an important role in educating the communities to participate in urban farming activities [90,91]. For example, in Kedah, problems such as land, capital, crop diseases, and access to agricultural equipment were the major problems faced by those involved in urban farming [92].

Focus should also be on how urban communities can be encouraged further to participate in UF activities. Society recognition, attitude, and social impact of urban farming are the top three considerations for individuals to participate in urban farming in Malaysia,

while value, economy, and personal reasons are factors that influence public acceptance of urban farming [86,93].

Stakeholders, especially academicians, can also play a very important role especially in protecting agricultural land by carrying out and suggesting solutions related to UF mechanisms [93]. Recommending comprehensive agricultural land preservation policies to the government may enhance the effectiveness and efficiency of any agricultural initiatives that are implemented. Additionally, the preparation and circulation of easy-to-follow designs or manuals for novices to start UF at home (starter-kits) will be of great help to communities which have little-to-no knowledge about agriculture.

Comprehensive implementation of a combination of the relevant initiatives suggested in this paper can contribute to better uptake and success of UF activities in the country. This will be a boon to the nation where food demand is constantly on the rise.

## 5. Conclusions

The COVID-19 pandemic outbreak resulted in severe health issues and, at the same time, led to severe food supply issues. The implementation of strict MCO policies within local and international borders, particularly during the early periods of the outbreak, to curb the spread of the virus, had severe adverse effects on the food supply chain. It was more severe particularly on the distribution of locally produced fresh food and imported food to urban areas, with disruption of food supplies for several communities in Malaysia [34]. The situation was especially bad during the implementation of the first related MCO, where, due to movement restrictions, fresh food supplies such as vegetables from local production areas ended up being dumped and never reached the intended local consumers.

Urban farming or home cultivation is thus considered the way forward to ensure better food quality and food security in Malaysia [81]. The beginnings of such a realisation are seen in the activities during the recent crisis. Movement and business restrictions during the pandemic caused many individuals to lose their jobs or to work from home and they instead spent their time doing home gardening. Further, out of necessity and to fill their time, city dwellers began growing their own vegetables in vacant areas in yards, verandas, or on rooftops, using pots. The number of locations and urban farming participation increased during the pandemic (refer to Tables 4 and 5).

However, will participation in UF increase or be sustained during the post-pandemic era when community life is back to normal and work at the office is as before the pandemic? Through social media research it was found that there was an increase in awareness and efforts in many municipalities in creating their respective community garden units during and after the pandemic. Even though the government and several Non-Governmental Organisations (NGO) have launched campaigns, organised programmes, and provided subsidies for urban farming to attract community participation, it appears to have been insufficient to attract the attention of the local urban community to become involved in farming activities prior to the pandemic. This may be due to limitations of knowledge, area, and space. Thus, to sustain and increase interest in urban farming, it is envisaged that the Government and NGOs will play an important role by promoting and conducting knowledge transfer activities on an ongoing basis in the post-pandemic era.

On 3 August 2021, the Ministry of Housing and Local Government (KPKT) launched the Urban Community Farm Policy (Dasar Kebun Komuniti Bandar (DKKB)), consistent with the role of KPKT as a leader in urban communities' prosperity and sustainable environments. The basic philosophy of this policy is to develop urban community gardens in an organised, systematic, organic, and sustainable manner. This policy is also viewed as a two-pronged approach to address the new challenges arising following the rapid development and spread of the COVID-19 disease and also support Malaysia's commitment to meet the Sustainable Development Goals (SDGs) 2030 by integrating elements of economic development, social development, and environmental conservation. DKKB's aspiration is to guide the urban community to develop the open spaces around their residential areas

with cultivation activities that can benefit the local community, contribute to sustainable development, and assist in facing the current diverse municipal challenges.

There are, however, a few important aspects that are not addressed in the DKKB, which will need to be looked into to further improve and strengthen this laudable initiative. Among the issues that need to be addressed are the organisation/management of community gardens/farms, product sales guarantees, i.e., implementation of contract farm agreements to ensure that the resultant products can be sold, and the method of profit distribution among members.

As the lack of suitable land areas for food production is a frequent issue faced by entrepreneurs and the private sector in the city area, the government may need to create Permanent Food Production Parks (TKPM) in urban areas, as has previously been done in rural areas. The TKPM will provide permanent sites for food production, especially fruits and vegetables. The TKPM can also play an important role in offering project sites for trainee entrepreneurs, for example, those from the Department of Agriculture Incubator Center. Urban TKPMs should also be provided with plant protection structures such as Urban BioDomes equipped with the latest technologies such as IR 4.0 and IoT. Vertical and hydroponic cultivation in rooftop areas can increase yields in limited space as seen with Singapore's introduction of Sky Green's vertical towers, which applies the vertical rooftop model for urban agriculture to feed their residents. Malaysia, which still uses conventional farming and basic hydroponics systems in urban and community gardens, will also need to look into implementation of such new technologies.

Urban agriculture does not fascinate the Malaysian youth mainly due to it being viewed as labour-intensive and not technology driven. The above and other measures put in place have to take into consideration how they can also pique the interest of the younger generation and encourage them to actively participate in urban agriculture programs that were previously seen to be dominated by older persons or pensioners. Incorporation of new technologies such as those suggested above and other new technologies available in the market today which make the tasks of urban farming easier and more productive may be key to continued and sustained interest of the urban community, both young and old. In this respect, educational institutions will also play an important role in producing a knowledgeable generation, especially in modern urban agriculture, with increased focus on new agricultural technologies.

In conclusion, due to the recent COVID-19 pandemic, which highlighted issues of food security in urban areas, the importance of urban agriculture as a measure to address food security during crisis situations has gained prominence in the Malaysian context. Collaboration between Government agencies, NGOs, and educational institutions will be needed to create and sustain awareness of the importance of this measure, particularly for urban food security. In the long run, partnerships between government agencies, the education sector, and the private sector are also necessary to develop modern urban agricultural technologies that will guarantee the urban community's interest in participation in urban agriculture activities. What is ultimately evident is that urban agriculture is here to stay and needs serious attention and participation of all stakeholders.

**Author Contributions:** R.M. and M.M. conceived and designed the manuscript. R.M. and M.M. wrote the paper. N.E.T., W.H.O. and Z.H. reviewed the manuscript's content flow (continuity) by stages. M.B.J. and A.A. corrected the final draft. All authors have read and agreed to the published version of the manuscript.

**Funding:** This research was funded by Universiti Malaysia Sabah Special Fund Grant (Skim Dana Khas: SDK0203-2020).

**Institutional Review Board Statement:** Not applicable.

**Informed Consent Statement:** Not applicable.

**Acknowledgments:** The authors are grateful for all the support and facilities provided by Universiti Malaysia Sabah; and the constructive suggestions and comments from colleagues and reviewers, which contributed immensely to the improvement of this paper. The authors would also like to thank the Urban Farming Division, Department of Agriculture Malaysia, for the statistical data contributed; and Anuar Tawe from the Urban Agriculture Unit, Department of Agriculture, Selangor, Malaysia; as well as Chong Khai Yeng and Muhammad Shahrul Khalid for their permission to use their photographs for this manuscript.

**Conflicts of Interest:** The authors declare no conflict of interest.

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
