# Peer review of "Ensuring Urban Food Security in Malaysia during the COVID-19 Pandemic—Is Urban Farming the Answer? A Review"

_sustainability, doi:10.3390/su14074155_

Round 1

Reviewer 1 Report

In the paper "Ensuring Urban Food Security in Malaysia During the COVID-19 Pandemic – Is Urban Farming the Answer?" A Review," the authors attempt to investigate the relationship between urban farming and food security during the Covid-19 pandemic. The authors have described Malaysia's current food security situation, as well as the state of urban farming, among other things. Unfortunately, due to serious flaws, the paper does not make a significant contribution to the body of literature. My comments on the manuscript are as follows:

Title

Remove the punctuation mark “:” from the title. Furthermore, I notice that the majority of the manuscript's content is unrelated to the title.

Abstract

Abstract should be more comprehensive and include more findings.

Introduction

The introduction is overly long and filled with irrelevant information. Please keep the introduction brief and relevant to urban farming and explain how its importance has grown since the COVID-19 pandemic. Also, please keep in mind that the food supply chains demonstrated remarkable resilience during the COVID-19 pandemic, and there are currently no global food shortages. Except for the early days of the pandemic, when panic buying caused some shortages, the overall situation of food supply chains has remained relatively stable, in contrast to the 2007-8 economic crisis. Therefore, I recommend that authors carefully link this issue of urban farming to COVID-19 while keeping it relevant to Malaysia. Remove any extraneous or irrelevant information.

Materials and Methods

This review paper's methodology is not that of a systematic review paper. Please adhere to the guidelines for composing a systematic review paper. Why did you use the Google Scholar search engine instead of the Web of Science? The keywords used are not all relevant to the topic under consideration.

Results and Discussion

 What is the source of Figure 1?

Much of the material under the Section 3.2 is irrelevant to the title of the heading.

What is the source of Figure 3 and Figure 4? Please mention the source of all Figures. Have you obtained the necessary permissions to reproduce other authors' work as such?

The paper devotes a significant amount of space to explaining what urban farming is and what types of urban farming exist. This is a well-known fact. Please keep your attention on your topic. It is unclear how you linked urban farming and the COVID-19. Despite the fact that this paper is about urban farming in Malaysia during COVID-19, I don't see much information that is relevant to the topic. Instead, the authors provided a lot of basic and unnecessary information about food security, COVID-19 progression, and the fundamentals of urban farming. Please revise the entire manuscript and indicate how many articles you discovered that are relevant to this review. I noticed that the authors only have 3 or 4 articles which are actually relevant to the urban farming.

What is the source of the tables in the manuscript? Have you gotten permission to use these tables?

What is the source of Figure 5? Is this authors’ own copyrighted material?

Overall, there is a severe lack of literature relevant to the issue under consideration in this article. The authors must revise it extensively in order to make it relevant to the topic and to identify themes upon which to build. There is currently insufficient analysis presented here. Instead, the authors have reported a lot of irrelevant information that adds no new knowledge.

Author Response

Dear Prof./Dr./ Mr./Ms.

Thank you for allowing me to submit a revised draft of my manuscript titled Ensuring Urban Food Security in Malaysia during the COVID-19 Pandemic – is Urban Farming the Answer? A Review. We appreciate the time and effort you have dedicated to providing your valuable feedback on my manuscript. We are grateful to the reviewers for their insightful comments on this paper. We have been able to incorporate changes to reflect most of the suggestions provided by the reviewers. Your comments immensely helped us in improving this paper.

Here is a point-by-point response to the reviewers’ comments and concerns.

Comment 1: Remove the punctuation mark “:” from the title. Majority of the manuscript's content is unrelated to the title.

Response:

Punctuation mark removed.

Ensuring Urban Food Security in Malaysia during the COVID-19 Pandemic – is Urban Farming the Answer? A Review

Comment 2: Abstract should be more comprehensive and include more findings.

Response: Abstract reconstructed to be more comprehensive and included with more findings (Refer to Abstract).

Comment 3: The introduction is overly long and filled with irrelevant information. Please keep the introduction brief and relevant to urban farming and explain how its importance has grown since the COVID-19 pandemic. Carefully link this issue of urban farming to COVID-19 while keeping it relevant to Malaysia. Remove any extraneous or irrelevant information.

Response: The introduction was reconstructed. Importance and relationship between urban farming and COVID-19 were added, some irrelevant information was removed. (Refer to Introduction, page 2)

Comment 4: This review paper's methodology is not that of a systematic review paper. Please adhere to the guidelines for composing a systematic review paper. Why did you use the Google Scholar search engine instead of the Web of Science? The keywords used are not all relevant to the topic under consideration.

Response: Review protocol used are RepOrting Standard for Systematic Evidence Syntheses (ROSES)- Methodology was reconstructed more details explaining the method used in this study (Refer to Methodology, page 3 - 5).

Comment 5: Source for Figure and table. Have you obtained the necessary permissions to reproduce other authors' work as such?

Response: Figures and tables have been put with the source/ citation.

Table were obtained from Department of Agriculture Malaysia, permission were ask through email and has already granted (Refer to Result, Table 4 and 5, line 503 and 513, page 14).

Comment 6: Much of the material under Section 3.2 is irrelevant to the title of the heading

Response: Much materials were included the economic status in Malaysia during the pandemic and government strategies to overcome the issues, such as urban farming. Title on 3.2.2 was changed to - Government Strategies and Initiative to Attract Interest of the Community to Participate in Urban Farming. (Refer to Results, Page 9, Line 301).

Comment 7: The paper devotes a significant amount of space to explaining what urban farming is and what types of urban farming exist. It is unclear how you linked urban farming and the COVID-19. Not much information that is relevant to the topic. Instead, the authors provided a lot of basic and unnecessary information about food security, COVID-19 progression, and the fundamentals of urban farming.

Response: The results discuss the effect of the pandemic on food distribution in Malaysia which cause food insecurity, especially during the first phase of the pandemic and when the implementation of MCO policy during COVID-19 cause insufficient protein and fibre intake.  . One of the government initiative to solve this problem is by launching community garden as the participant in an urban farming increase during the pandemic. Urban Community Garden Policy was launched in 2021 to encourage urban communities involved in urban farming activities by giving them guidelines in the policy. Refer to Result (Table 4 and 5, line 503 and 513, page 14). (Page 8-9, Line 295, 297) (Page 9, Line 350).

Reviewer 2 Report

I like what the authors are trying to achieve in this paper,  minor changes:
1.    It's not clear to me the connection between COVID-19 and the Urban Food Security in Malaysia especially in a review paper including Journal articles from 1994 to 2021. The authors have to justify better the role of the recent pandemic in food security. 
2.    The authors should not use the first person in scientific work (e.g. Introduction: line 98).
3.    It’s not very clear how the authors selected the included references. A review analysis model (e.g. PRIMA or else) will be very useful for Analyzing Place-Related Information 
4.    The conclusions section is very poor. The authors should justify better their arguments and include policy implications and also some limitations of their research. 

Author Response

Dear Prof./Dr./ Mr./Ms.

Thank you for allowing me to submit a revised draft of my manuscript titled Ensuring Urban Food Security in Malaysia during the COVID-19 Pandemic – is Urban Farming the Answer? A Review. We appreciate the time and effort you have dedicated to providing your valuable feedback on my manuscript. We are grateful to the reviewers for their insightful comments on this paper. We have been able to incorporate changes to reflect most of the suggestions provided by the reviewers. Your comments immensely helped us in improving this paper.

Here is a point-by-point response to the reviewers’ comments and concerns.

Comment 1: It’s not very clear how the authors selected the included references. A review analysis model (e.g. PRIMA or else) will be very useful for Analyzing Place-Related information 

Response: Review protocol used are RepOrting Standard for Systematic Evidence Syntheses (ROSES)- Methodology was reconstructed more details explaining the method used in this study (Refer to methodology, page 3 - 5).

Comment 2: The connection between COVID-19 and the Urban Food Security in Malaysia is not clear, especially in a review paper including Journal articles from 1994 to 2021. The authors have to justify better the role of the recent pandemic in food security. 

Response: At the beginning pandemic, food availability was disturbed due to the MCO policy in Malaysia cause the participation of Malaysians in Urban Farming increased largely in 2020 compared to the year before, here the community use urban farming to secure their food source also to gain side income during the pandemic. Refer to (page 6, sub- 3.1, para 5), (page 11, sub-3.3.2, para 2), (page 14, 15, sub- 3.4.1, para 7, 9), (Page 16, Sub- 3.4.2, para 3), (page 17, sub- 3.5, para 6)

Participation in urban farming by the community is very important to ensure the community are able to secure the food source. The younger generation seems more flexible in adapting and accepting, and able to relate UF to food security. Here, the participation by the younger generation can ensure food security during the pandemic, even after that. (Refer to the discussion, page 19, para 3, 4).

Comment 3: The authors should not use the first person in scientific work (e.g. Introduction: line 98).

Response: The introduction was reconstructed. Importance and relationship between urban farming and COVID-19 were added, some irrelevant information was removed. (Refer to Introduction, page 2)

Comment 4: The conclusions section is very poor. The authors should justify better their arguments and include policy implications and also some limitations of their research

Response: All paragraphs of the conclusion done reconstructed. (Refer to Conclusion, page 18)

Round 2

Reviewer 1 Report

The authors have revised the paper correctly. The manuscript can be accepted for publication.